

# Low self-concept in poor readers: prevalence, heterogeneity, and risk

Genevieve McArthur, Anne Castles, Saskia Kohnen and Erin Banales

Department of Cognitive Science, ARC Centre of Excellence in Cognition and its Disorders, Macquarie University, Sydney, New South Wales, Australia

## ABSTRACT

There is evidence that poor readers are at increased risk for various types of low self-concept—particularly academic self-concept. However, this evidence ignores the heterogeneous nature of poor readers, and hence the likelihood that not all poor readers have low self-concept. The aim of this study was to better understand *which types of poor readers have low self-concept.* We tested 77 children with poor reading for their age for four types of self-concept, four types of reading, three types of spoken language, and two types of attention. We found that poor readers with poor attention had low academic self-concept, while poor readers with poor spoken language had low general self-concept in addition to low academic self-concept. In contrast, poor readers with typical spoken language and attention did not have low self-concept of any type. We also discovered that academic self-concept was reliably associated with reading and receptive spoken vocabulary, and that general self-concept was reliably associated with spoken vocabulary. These outcomes suggest that poor readers with multiple impairments in reading, language, and attention are at higher risk for low academic and general self-concept, and hence need to be assessed for self-concept in clinical practice. Our results also highlight the need for further investigation into the heterogeneous nature of self-concept in poor readers.

# INTRODUCTION

Sixteen percent of children have reading skills that fall below the average range for their age, and 5 percent of children have significant and severe reading difficulties (*Shaywitz et al., 1992*). These children's reading difficulties vary in aetiology. Some children struggle to learn new skills in general ("general learning difficulty"), while others struggle to learn to read despite adequate instruction and the ability to learn new skills in general ("development dyslexia"; "specific reading disability"). These children's reading difficulties also vary in type. Some poor readers have problems with learning to read words accurately (reading accuracy), some with learning to read words fluently (reading fluency), some with understanding what they read (reading comprehension), and many with different combinations of these problems (*Stuart & Stainthorpe, 2016*).

We have known for quite some time that poor reading puts children at higher risk for academic failure (*Herbers et al., 2012*; *Smart et al., 2001*). However, we are only just

Corresponding author
Genevieve McArthur,
genevieve.mcarthur@mq.edu.au

starting to understand how poor reading affects children's emotional health. In the current study, we focus on poor readers' self-concept, which can be defined as a "person's perceptions of him- or herself. These perceptions are formed through experience with and perceptions of one's environment. They are influenced especially by evaluations by significant others, reinforcements, and attributions for one's own behaviour" (*Marsh & Shavelson, 1985*; p. 107). Positive self-concept is associated with numerous important facets of life, such as academic achievement (*Marsh & Craven, 2006*), economic success and health (*Organisation for Economic Cooperation Development, 2003*), emotional adjustment (*Donahue et al., 1993*), coping (*Shirk, 1988*), and happiness (*Harter, 1990*).

Self-concept typically refers to a person's perceptions of her- or himself in a particular domain (e.g., academic, social, parent-home, or physical; (*Cole et al., 2001*; *Harter, Whitesell & Junkin, 1998*; *Marsh & Seaton, 2013*). This contrasts with self-esteem, which has been defined as "one's global sense of well-being as a person" (p. 148, *Zeleke, 2004*). In the context of an academic domain such as reading, it is important to discriminate between different domains of self-concept, as well as between self-concept and self-esteem, since academic achievement appears to have a reciprocal relationship with some domains (i.e., academic self-concept) and not others (i.e., non-academic domains), and no reciprocal relationship with self-esteem at all (*Marsh & Martin, 2011*).

Around two-dozen studies have tested poor readers for at least one domain of self-concept, including academic self-concept (*Alexander-Passe, 2006*; *Terras, Thompson & Minnis, 2009*), social self-concept (e.g., *Martínez & Semrud-Clikeman, 2004*; *Snowling, Muter & Carroll, 2007*), athletic self-concept (e.g., *Boetsch, Green & Pennington, 1996*; *Frederickson & Jacobs, 2001*), physical self-concept (e.g., *Humphrey & Mullins, 2002*; *Casey et al., 1992*), behavioural self-concept (e.g., *Frederickson & Jacobs, 2001*; *Murray, 1978*), parental self-concept (e.g., *Thomson & Hartley, 1980*; *Westervelt et al., 1998*), reading self-concept (e.g., *Bull, 2007*; *Morgan et al., 2008*), and practical self-concept (e.g., *Polychroni, Koukoura & Anagnostou, 2006*). Of these domains, it is academic self-concept that appears to be most reliably impaired in poor readers (*Alexander-Passe, 2006*; *Boetsch, Green & Pennington, 1996*; *Casey et al., 1992*; *Frederickson & Jacobs, 2001*; *Humphrey & Mullins, 2002*; *Murray, 1978*; *Snowling, Muter & Carroll, 2007*; *Terras, Thompson & Minnis, 2009*; *Thomson & Hartley, 1980*; *Westervelt et al., 1998*). For example, *Taylor, Hulme, & Welsh (2010)* tested twenty-six 8- to 12-year-old poor readers in mainstream classrooms for their self-concept in scholastic competence (academic self-concept), social acceptance, athletic competence, physical appearance, and behavioural conduct. Compared to 23 children with no learning disability, the poor readers only had poor scores for academic self-concept. Similarly, *Snowling, Muter & Carroll (2007)* tested twenty-one 12- to 13-year-old poor readers for their perceived scholastic competence (academic self-concept), social competence, and athletic competence. They too found that, compared to age-matched typical readers ($N = 17$), poor readers scored poorly on academic self-concept alone. Thus, studies of self-concept in poor readers to date have produced mixed findings, suggesting that poor readers are at increased risk for various types of low self-concept, particularly low academic self-concept.

Unfortunately, these studies have not considered individual differences in poor readers' self-concept. This is surprising given that (1) poor readers are known to be a highly heterogeneous population, and (2) we have yet to discover a single impairment—emotional, genetic, neurological, cognitive, environmental, or behavioural—that is present in all poor readers. It is therefore likely—if not a surety—that some poor readers do not have low self-concept. Yet, to our knowledge, no study has attempted to understand which types of poor readers are more likely to have poor self-concept.

There are three areas of cognition that poor readers reliably vary in type: their reading, their language, and their attention. Regarding their reading, and as mentioned above, some poor readers have problems with reading accuracy, some with reading fluency, some with reading comprehension, and many have different combinations of these problems. To complicate things further, within the domain of reading accuracy alone, some children have problems with learning to read words via phonological recoding (the ability to decode words using the letter-sound rules), some via visual word recognition (i.e., the ability to recognise whole words from a mental store, or lexicon, of written words), and some via both phonological recoding and sight word reading (*McArthur et al., 2013a*; *McArthur et al., 2013b*; *Stuart & Stainthorpe, 2016*). Thus, poor readers vary considerably in the nature of their reading impairments. This variance can be captured via tests of phonological recoding accuracy, visual word recognition accuracy, reading fluency, and reading comprehension.

Regarding spoken language, there is abundant evidence that some (but not necessarily all) poor readers have concomitant impairments in their spoken language abilities (*Bishop & Snowling, 2004*; *Catts et al., 2005*; *Eisenmajer, Ross & Pratt, 2005*; *Fraser, Goswami & Conti-Ramsden, 2010*; *McArthur et al., 2000*; *Rispens & Been, 2007*). After reviewing this evidence, *Bishop & Snowling (2004)* proposed that language impairments in poor readers might be categorized as either phonological in nature (e.g., phonological representations, phonological segmentation, phonological memory) or non-phonological in nature (e.g., semantics and syntax). They suggested that phonological impairments in poor readers might be indexed using nonword repetition tasks, and that non-phonological impairments could be indexed using receptive and expressive measures of vocabulary knowledge.

Regarding attention, previous studies have reported that poor readers, as a group, are more likely to have inattention or hyperactivity than typical readers (e.g., *Gilger, Pennington & DeFries, 1992*; *Shaywitz, Fletcher & Shaywitz, 1995*; *Willcutt & Pennington, 2000*). Similarly, there is evidence that some (but not all) children diagnosed with attention deficit disorder (i.e., around 50%) have poor reading relative to children with typical attention (*Dykman & Ackerman, 1991*; *Semrud-Clikeman et al., 1992*). These studies indicate that, poor readers vary in their levels of attention. These differences can be measured via tests of inattention and hyperactivity.

In sum, the evidence to date suggests that poor readers may be at increased risk for various types of low self-concept—most particularly academic self-concept. However, this evidence overlooks the fact that poor readers represent a highly varied population—particularly in terms of their reading, their spoken language, and their attention. It is therefore likely that

only some poor readers have low self-concept. *The specific aim of this study was to better understand which types of poor readers have low self-concept* by conducting three analyses. In the first, we estimated the prevalence of low scores in poor readers for four types of low concept (academic, general, home, and social). From the existing evidence discussed above, we predicted that our sample of poor readers would have *a disproportionately high number of low scores for their academic self-concept but not their general, home, or social self-concept.*

In the second analysis, we divided our poor readers into groups that had either (1) poor reading alone, (2) poor reading and poor spoken language, (3) poor reading and poor attention, or (4) poor reading, poor spoken language, and poor attention. We compared these groups for scores on the four self-concept scales. Previous research has found low self-concept in people with specific language impairment (*Carroll & Iles, 2006*; *Carroll et al., 2005*; *Lindsay & Dockrell, 2000*) and people with poor attention (*Maughan & Carroll, 2006*; *Treuting & Hinshaw, 2001*). There is also evidence that poor readers with concomitant problems with spoken language or attention are at higher risk for more severe cognitive deficits than poor readers with typical language and attention (*Eisenmajer, Ross & Pratt, 2005*; *Fraser, Goswami & Conti-Ramsden, 2010*; *McArthur & Castles, 2013*; *McArthur & Hogben, 2001*; *Willcutt et al., 2001*). Combining this evidence, we tentatively predicted that *poor readers with two comorbid problems (i.e., poor language and poor attention) would have more problems with self-concept than poor readers with one comorbid problem (i.e., poor language or poor attention), who would have more problems with self-concept than poor readers with no problems with language and attention.*

In the final analysis, we used correlation coefficients to determine which types of reading ability (phonological recoding accuracy, visual word recognition accuracy, reading fluency, reading comprehension), spoken language ability (phonological processing, spoken vocabulary knowledge), or attention (inattention, hyperactivity) might be reliably related to specific types of self-concept. To our knowledge, the specific relationships between different types of reading, language, attention and self-concept have never been examined before, and hence the outcomes of this analysis were necessarily exploratory.

## METHODS

### Informed consent and ethics approval

The University Human Research Ethics Committee approved the methods of this study (5201200852). The parents of all children gave informed written consent for their child to participate in the study. In addition, children gave their informed verbal consent to participate in the study.

### Participants

This study recruited 77 children with poor reading from the general community. The study was advertised via newspaper advertisements, via a Kids' Science Club, and via letters to schools. All children were aged from 9 to 12 years since this was the appropriate age-range for the self-concept subtests (see Self-concept Measures below). In addition, they scored at least 1 *SD* below the age-expected level for either phonological recoding

**Table 1  Sample characteristics.**

|  | M | SD | Min | Max |
|---|---|---|---|---|
| Age (years) | 10.46 | 1.01 | 9.00 | 12.50 |
| Sex (1 = female; 2 = male) | 1.42 | 0.50 | 1 | 2 |
| Nonverbal IQ (StS) | 98.52 | 15.77 | 64 | 138 |
| **Reading** |  |  |  |  |
| Phonological recoding (z) | −1.73 | 0.66 | −3.09 | 0.66 |
| Visual word recognition (z) | −1.36 | 0.74 | −3.09 | 0.94 |
| Reading fluency (z) | −1.00 | 0.82 | −3.07 | 0.67 |
| Reading comprehension (z) | −0.86 | 1.01 | −3.09 | 1.22 |
| **Self-concept** |  |  |  |  |
| Academic (ScS) | 8.53 | 2.84 | 1 | 13 |
| General (ScS) | 8.23 | 2.98 | 2 | 14 |
| Home (ScS) | 10.31 | 2.62 | 2 | 13 |
| Social (ScS) | 9.62 | 3.06 | 1 | 13 |
| **Spoken language** |  |  |  |  |
| Phonological processing (ScS) | 9.14 | 2.19 | 1 | 13 |
| Receptive vocabulary (StS) | 97.52 | 11.69 | 73 | 130 |
| Expressive vocabulary (ScS) | 7.73 | 2.22 | 3 | 16 |
| **Attention** |  |  |  |  |
| Inattention (/3) | 1.28 | .68 | .11 | 2.78 |
| Inattention (z) | .56 | .82 | −.98 | 2.15 |
| Hyperactivity (/3) | .59 | .56 | .00 | 2.22 |
| Hyperactivity (z) | −0.07 | .93 | −1.07 | 2.65 |

**Notes.**

Mean (*M*), Standard Deviation (*SD*), Minimum (Min) and Maximum (Max) Standard Scores (*StS*; $M = 100, SD = 15$), Scaled Scores (*ScS*; $M = 10, SD = 3$) and *Z* Scores ($z$; $M = 0, SD = 1$) Produced by our Sample of Poor Readers ($N = 77$).

or visual word recognition (see Screening Tests below); spoke English as their first language; and had no history of neurological or sensory impairment, as indicated on a background questionnaire. While we measured children's non-verbal IQ for information (see Screening Measures and Table 1), we did not exclude children based on non-verbal IQ scores since nonverbal IQ does not appear to predict the ability to learn to read (*Gresham & Vellutino, 2010*; *Siegel, 1989*). The children attended a variety of public and private schools. Given the absence of strict regulations about how reading should be taught in the school system, our sample would have received a variety of reading instructions, ranging from predominantly phonics instruction, to a mixture of phonics and sight word instruction, to "whole-word" instruction, which focuses primarily on meaning and reading strategies.

In line with previous studies (e.g., *Alexander-Passe, 2006*; *Bull, 2007*; *Terras, Thompson & Minnis, 2009*; *Westervelt et al., 1998*), this study did not recruit a control group because all tests were normed on large samples of typically-developing children who would have produced more reliable data than an aged-matched, sized-matched control group recruited for this study (i.e., $N = 77$ across 9-, 10-, 11-, and 12-year-old age groups). Further, in line with most standardised assessment, this study considered scores from $-1$ *SD* to $+1$

*SD* as representative of the average range. Standard scores, scaled scores, or *z* scores below −1, 85, or 7 (respectively)—which represent the lowest 15.9% of scores in a normal distribution-were considered to be "low" (with regards to self-concept) or "poor" (with regards to reading or language). In addition, we considered *z* scores from −1 to −0.5, standard scores from 85 to 92, and scaled scores from 7 to 8 to be "low-average"; and we considered *z* scores from −0.51 to +1, standard scores from 86 to 115, and scaled scores from 9 to 13 to be "average".

The majority of children produced complete data sets. One child's parents did not complete the inattention and hyperactivity questionnaire, and hence that child could not be assigned to a group for the second analysis, and they did not contribute to correlation coefficients including inattention and hyperactivity in the third analysis. Six children did not complete the nonverbal intelligence test. Since this test was used to get a general sense of the sample, and was not included in any of the analyses in this study, the absence of this data had little impact on the validity of the outcomes.

Table 1 provides descriptive statistics for the test scores of the 77 poor readers in this study. These statistics indicated that this sample of children had, on average, poor scores for phonological recoding and visual word recognition; low-average scores for reading fluency and reading comprehension, academic and general self-concept, as well as expressive vocabulary; and average scores for nonverbal IQ, home and social self-concept, as well as phonological processing and receptive vocabulary (see sections below for descriptions of tests). Their scores for inattention and hyperactivity fell well below cut-offs for clinical significance (1.78 for and 1.44, respectively). Thus, overall, our sample of poor readers had marked, but not unusually severe, reading problems. Examination of *SD*, minimum and maximum values further indicated that some poor readers had concomitant deficits in their spoken language and attention. Such a sample is representative of English-speaking poor readers found in mainstream primary-school classrooms in the UK, US, and Australia.

## Procedure

Each child in our sample was invited to the University to complete the Screening, Self-concept, Reading, and Language Measures (see below) individually in a quiet testing room. The measures represented part of a larger test battery that took 2–3 hours to complete, depending upon the child's age, ability, and personality. The tests were administered in a fixed order that separated longer tests (e.g., non-verbal IQ) with shorter tests (reading fluency). Children were given positive reinforcement and encouragement throughout the testing session regardless of their level of achievement. At the completion of each test, a child was given to sticker to put on a progress chart to help them track their achievement and progress. They were also given numerous breaks, during which they played games with the tester or had a snack (approved by parents). With parental permission, children were rewarded for their efforts with $30. Parents completed the Attention Measures at home or at the University while their child completed their tests. Their travel costs were reimbursed with $10–15, depending upon distance travelled to the University.

## Screening measures
### Phonological recoding and visual word recognition

We tested these two skills using the Castles and Coltheart 2nd Edition (CC2) Nonword subtest and the CC2 Irregular Word subtest, respectively (*Castles et al., 2009*). The CC2 comprises three lists: 40 nonwords (e.g., GRENTY), 40 irregular words (e.g., TOMB), and 40 regular words (e.g. STENCH). The stimuli within each list are presented in order of increasing difficulty in an inter-mixed fashion (e.g., regular word 1, irregular word 1, nonword 1, nonword 2, irregular word 2, regular word 2, irregular word 3, regular word 3, nonword 3, and so on). Stimuli within each list are presented until a child makes five consecutive errors within that list. Items in other lists continue to be presented until a child makes five consecutive errors within a list, or they reach the end of the test. Scores for each list are *z* scores with a mean of 0 and *SD* of 1. Internal consistency for this test is .94 (*Moore et al., 2012*).

### Nonverbal intelligence

We tested nonverbal intelligence using the Matrices subtest of the Kaufman Brief Intelligence Test 2nd Edition (KBIT-2; *Kaufman & Kaufman, 2004*). In each trial, children are shown an incomplete picture matrix, and asked to select the missing portion from four or six possible options. Scores are standard scores with a mean of 100 and an *SD* of 15. The split-half reliability for this subtest is .80–.90, and the test-retest reliability is .83.

## Self-concept measures

Self-concept was measured with the Culture Fair Self Esteem Inventory (3rd Edition) that was designed for children aged 9- to 12-years-old (CFSEI-3; *Battle, 2002*). It comprises four subtests, one each for academic, general, home, and social self-concept. The CSFEI-3 comprises 64 statements, which were each read aloud to the poor readers in this study. Each statement referred to a child's self-concept in the academic domain (10 items; e.g., "I am satisfied with my schoolwork"), general domain (14 items; e.g., "Most boys and girls are better at doing things than me"), home domain (12 items; e.g., "My family thinks I am important"), and social domain (18 items; e.g., "Boys and girls like to play with me"). Each domain was measured using scaled scores that had a mean of 10 and standard deviation of 3. Mean internal consistency and time sampling reliability coefficients for the four domains range from .72 to .98.

## Reading measures
### Reading accuracy

As described under Screening Measures above, we tested these two skills using the CC2 Nonword subtest and the CC2 Irregular Word subtest, respectively (*Castles et al., 2009*).

### Reading fluency

We measured word reading fluency using the Sight Word subtest of the widely-used Test of Word Reading Efficiency (TOWRE; *Torgesen, Wagner & Rashotte, 1999*). In this subtest, children are asked to read a mix of regular and irregular words as quickly as they can within 45 s. The TOWRE was designed to produce standard scores with a mean of 100 and standard deviation of 15. In this study, we converted these standard scores into *z* scores

(with a mean of 0 and *SD* of 1) to make the reading fluency scores directly comparable to the reading accuracy and reading comprehension *z* scores. The mean parallel form and test-retest reliability coefficients for the TOWRE exceed .90.

### Reading comprehension

This was assessed using the Test of Everyday Reading Comprehension (TERC; *McArthur et al., 2013a*; *McArthur et al., 2013b*). This includes 10 "everyday" reading stimuli, such as a text message or a medicine label. For each stimulus, children are asked two literal questions about the information in the text. Scores are *z* scores with a mean of 0 and standard deviation of 1. The parallel-form and inter-rater reliability of the TERC is .86 and .99, respectively. The validity of the TERC is supported by a study by *Wheldall & McMurtry (2014)* who report that TERC scores in poor readers are strongly correlated ($r = .71$ $p < .05$) with the widely-used reading comprehension subtest of the Neale Analysis of Reading Ability (*Neale, 1999*), which uses short stories as text stimuli, rather then everyday text stimuli.

## Spoken language measures
### Phonological processing

We assessed phonological processing with a nonword repetition test: the standardized Neuropsychological Assessment (NEPSY) Repeating Nonwords subtest (*Korkman, Kirk & Kemp, 1998*). In this test, children are asked to repeat 13 nonsense words that increase in length (e.g., "ba-fee" to "skri-flu-na-fliss-trop"). Scores are scaled scores with a mean of 10 and a *SD* of 3. The mean test-retest reliability of this subtest for school-aged children is .74 (*Brooks, Sherman & Strauss, 2010*).

### Receptive vocabulary

To test receptive vocabulary, we used the Peabody Picture Vocabulary Test 4th Edition (PPVT-4; *Dunn & Dunn, 2007*). For each item, children are shown four pictures and asked to point to the picture that is named by the tester. Scores are standard scores with a mean of 100 and an *SD* of 15. The test-retest, split-half, and parallel-form reliability coefficients of this test are .92–.96, .89–.97, and .87–.93, respectively.

### Expressive vocabulary

We tested expressive vocabulary using the standardized Picture Naming subtest from the Assessment of Comprehension and Expression 6–11 (ACE; *Adams et al., 2001*). Children are asked to name the object in each of 25 pictures. Scores are scaled scores with a mean of 10 and *SD* of 3.

## Attention measures
### Inattention

This was measured with the Swanson, Nolan, and Pelham (4th Edition; SNAP-4; *Swanson et al., 1983*), which is a parent questionnaire that comprises nine descriptions of a child's behaviour that index "inattention". Mean and *SD* data for girls and boys aged 5–11 years published by *Bussing et al. (2008)* were used to calculate age- and sex-appropriate *z* scores for each child, which had a mean of 0 and *SD* of 1. Internal reliability for the SNAP overall is

.94 (parents) and .97 (teachers; *Bussing et al., 2008*). The raw score for clinical significance is 1.78.

### Hyperactivity

This was also measured with the SNAP-4, which includes nine descriptions that index "hyperactivity-impulsivity". Again, mean and *SD* data for girls and boys aged 5 to 11 years, published by *Bussing et al. (2008)*, were used to calculate age- and sex-appropriate *z* scores for each child. The raw score for clinical significance is 1.44.

### Data analysis

The data analysis comprised three steps. As outlined in the Introduction, the aim of the first step was to determine if poor readers had a *disproportionately high number of low scores for their academic self-concept but not their general, home, or social self-concept.* Prevalence of low scores was measured as the percentage of scores lower than $-1$ *SD* (ScS = 7) on each self-concept scale. To test if prevalence of low scores was atypical for a normal distribution, we used one-sample *t*-tests to compare each set of self-concept scores to the mean (*M*) expected for a typical population (ScS = 10). We used one-tailed tests of significance to determine if low scores were statistically reliable ($p < .05$), rather than typical two-tailed tests of significance, because we had a clear prediction about direction of scores (i.e., low in poor readers).

The outcomes of the first step in the analysis suggested that distributions of scores for at least two of the self-concept measures (academic and general) might differ significantly from a normal distribution. This was confirmed by Levene Tests for Normality, which revealed that, the distribution of scores for 10 of the 13 variables in this study differed significantly from a normal distribution. Thus, non-parametric statistics were used in the second and third analyses.

The aim of the second step of the analysis was test if *poor readers with two comorbid problems have more problems with self-concept than poor readers with one comorbid problem, who would have more problems with self concept with poor readers with poor reading alone.* We divided poor readers into four groups who either had poor reading alone, poor reading and poor spoken language, poor reading and poor attention, or poor reading and poor spoken language and poor attention. After ensuring that the four groups differed only in their spoken language and attention (i.e., not their reading), we used One-Sample Wilcoxon Signed Rank Tests and Kruskal–Wallis Tests to compare their median scores for four types of self-concept (academic, general, home, and social) to a typical population and to each other, respectively. Again, because we had a clear prediction about direction of scores (i.e., low in poor readers), we used one-tailed tests of significance ($p < .05$).

In the third step of the analysis, we used Spearman Rho Rank Correlation Coefficients to determine which types of reading ability (phonological recoding accuracy, visual word recognition accuracy, reading fluency, reading comprehension), spoken language ability (phonological processing, spoken vocabulary knowledge), or attention (inattention, hyperactivity) were reliably related to different domains of self-concept (academic, general,

home, and social). As outlined in the Introduction, this analysis was exploratory. We made no predictions about direction of outcomes and so used two-tailed tests of significance ($p < .05$).

## RESULTS

### Step 1: prevalence of low academic, general, social, and home self-concept scores in poor readers

Compared to a typical population in which 15.9% of scores fall below $-1$ $SD$, our sample of poor readers had an unusually high rate of low scores for academic self-concept ($19/77 = 25\%$; $M = 8.53$; $SD = 2.84$; $t(76) = 4.14$, $p < .001$) and general self-concept ($23/77 = 30\%$; $M = 8.23$; $SD = 2.98$; $t(76) = 5.20$, $p < .001$). In contrast, the percentage of poor readers with low scores for home self-concept ($7/77 = 9\%$; $M = 10.31$; $SD = 2.62$; $t(76) = 1.04$, $p = .15$) and social self-concept ($15/77 = 19\%$; $M = 9.62$; $SD = 3.06$; $t(76) = 1.08$, $p = .11$) was not different to that expected for a typical population.

### Step 2: comparing self-concept in poor readers, with and without poor language and/or attention

We divided our sample of poor readers into four groups: those with poor scores (i.e., lower than 1 $SD$) on at least one measure of (1) reading, but no measure of language or attention (Reading group), (2) reading and spoken language, but no measure of attention (Reading + Language group), (3) reading and attention, but no measure of language (Reading + Attention group), or (4) reading and spoken language and attention (Reading + Language + Attention group). As outlined above, before comparing these four groups for self-concept scores, we tested whether the poor readers in each group differed in their reading profiles (i.e., in addition to differing in their language and attention profiles). Figure 1 shows the median reading scores (with upper and lower quartiles) of the four groups. The horizontal line in this figure represents the median level expected for each child's age (i.e., a $z$ score of 0). Any median (short black horizontal line) within a patterned box that fell below the horizontal line indicates a distribution of scores that was reliably poorer than the median level expected for their age.

The four groups had similar medians for phonological recoding and visual word recognition. The two groups with poor spoken language tended to have lower median scores on the two reading tests that were reliably correlated with spoken receptive vocabulary (reading comprehension: $r_s = 42$, $p < .001$; reading fluency: $r_s = .38$, $p = .001$) and spoken expressive vocabulary (reading comprehension $r_s = .33$, $p = .002$; reading fluency: $r_s = .29$, $p = .005$). Nevertheless, Kruskal–Wallis tests revealed that there was no significant difference between the four groups on any of the reading tests: phonological recoding ($H = 3.83$, $p = .14$), visual word recognition ($H = 4.56$, $p = .10$), reading fluency ($H = 4.52$, $p = .10$), and reading comprehension ($H = 3.93$, $p = .14$).

Having established that the four groups did not differ reliably in their reading profiles, we turned to their self-concept. Figure 2 shows the median self-concept scores (with upper and lower quartiles) of the four groups defined above, along with the median level expected for each child's age (i.e., a scaled score of 10). One-Sample Wilcoxon Signed Rank Tests
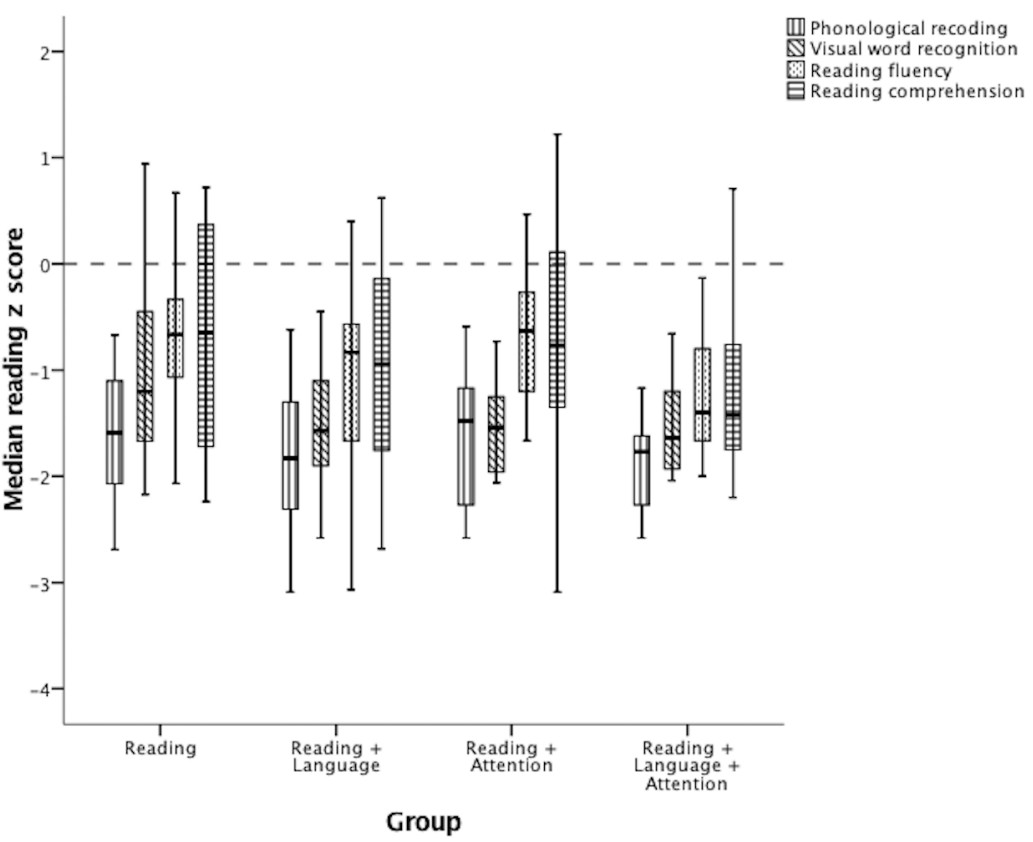

**Figure 1 Median reading scores (with upper and lower quartiles) of the four groups.** The horizontal line represents the median level expected for each child's age (i.e., a *z* score of 0). Any median (short black horizontal line) within a patterned box that falls below the horizontal line suggests a statistically reliable low median score for that measure for that group.

were used to compare each distribution of self-concept scores in each group to the median score expected for a child's age (i.e., ScS = 10). The associated medians (*Md*), means (*M*), standard deviations (*SD*), Wilcoxon Signed Rank Test values (*Z*), *p* values (*p*), and effect size values (*r*) are shown in Table 2 for each group for each type of self-concept. Statistically significant differences are marked in grey. Effect size *r* values of 0.1, 0.3, and 0.5 were considered small, moderate, and large, respectively.

The results revealed that (1) in the Reading Group, no type of self-concept fell significantly below the age-expected median (i.e., 10); (2) in the Reading + Language Group, academic self-concept and general self-concept fell significantly below the age-expected median; (3) in the Reading + Attention Group, academic self-concept fell significantly below the age-expected median; (4) in the Reading + Language + Attention group, academic and general self-concept fell below the age-expected median. All statistically significant effects were large in size. It is noteworthy that the Reading group had higher-than-expected median scores for home and social self-concept. Because we used one-tailed significance tests to detect low scores, we cannot speak to the reliability of the higher home and social self-concept scores in this study.

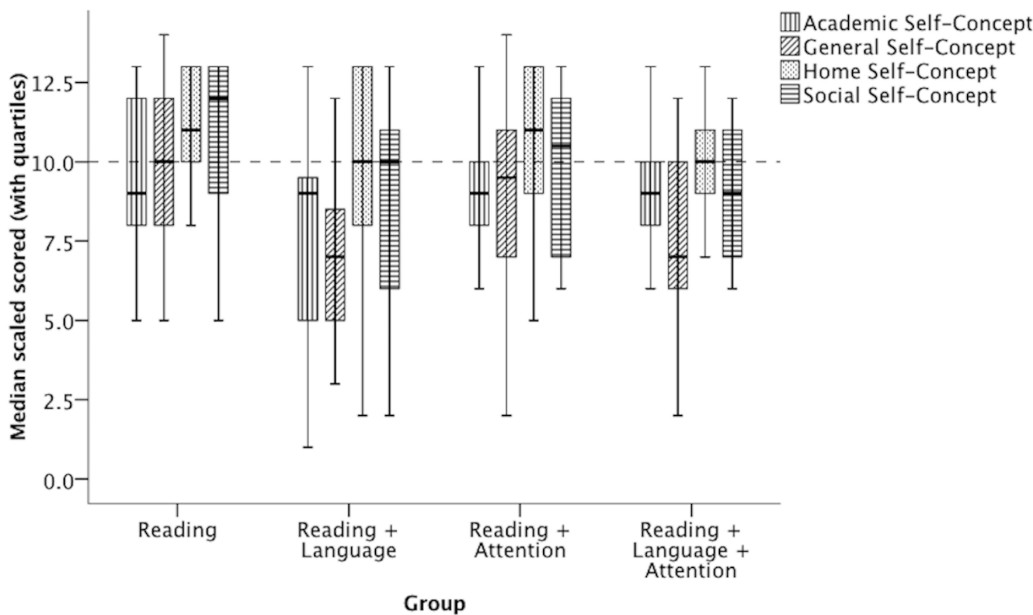

**Figure 2** **Median self-concept scores (with upper and lower quartiles) of the four groups.** The horizontal line represents the median level expected for each child's age (i.e., a scaled score of 10). Any median within a patterned box that falls below the horizontal line suggests a distribution of scores that might be statistically significantly different to the mean level expected for age.

We also used independent samples Kruskal–Wallis Tests to compare across groups for each type of self-concept. There was no significant difference between the groups for academic ($H = 2.55, p = .24$) or home ($H = 4.47, p = .11$) self-concept. There was a significant difference between groups for general self-concept ($H = 12.72, p = .002$). Post-hoc Mann–Whitney $U$ tests revealed that the two groups with language impairment had lower general self-concept than the Reading group (Reading + Language Group: $Z = -3.51, p < .001$; Reading + Language + Attention: $Z = -1.96, p = .002$). Thus, the presence of low general self-concept appeared to be associated with the presence of poor spoken language rather than poor reading or poor attention.

The Kruskal–Wallis Tests also revealed a significant difference between groups for social self-concept ($H = 8.60, p = .02$). Post-hoc Mann–Whitney $U$ tests revealed that the Reading Group had significantly higher scores than the Reading + Language group ($Z = 2.32, p = .01$) and the Reading + Language + Attention group ($Z = 2.52, p = .005$). Since the latter two groups had near-median social self-concept scores for their age, these group effects were driven by the unusually high scores of the Reading group, the reliability of which could not be ascertained by this study due to the use of one-tailed significance tests designed to detect low scores. Thus, this group effect was not considered further.

### Step 3: the relationship between poor readers' self-concept, reading, spoken language, and attention

Since Step 2 revealed that some types of poor readers have reliably poor scores for academic self-concept (i.e., if they have poor spoken language or attention) or for general self-concept

**Table 2  Poor readers' self-concept scores.**

| Group | Statistics | Self-concept | | | |
|---|---|---|---|---|---|
| | | Academic | General | Home | Social |
| Reading N = 25 | M | 9.32 | 9.76 | 11.16 | 10.64 |
| | SD | 2.27 | 2.57 | 3.00 | 3.05 |
| | Md | 9.00 | 10.00 | 11.00 | 12.00 |
| | Z | −1.11 | −0.39 | 2.82 | 1.68 |
| | p | .14 | .34 | .002 | .04 |
| | r | .22 | .08 | .56 | .34 |
| Reading + Language N = 24 | M | 7.63 | 6.88 | 9.58 | 8.83 |
| | SD | 3.56 | 2.38 | 3.22 | 3.45 |
| | Md | 9.00 | 7.00 | 10.00 | 10.00 |
| | Z | −2.69 | −3.93 | −0.13 | −1.27 |
| | p | .004 | <.001 | .44 | .10 |
| | r | .55 | 0.80 | 0.03 | 0.26 |
| Reading + Attention N = 14 | M | 8.57 | 8.64 | 10.14 | 10.14 |
| | SD | 2.71 | 3.43 | 2.77 | 2.63 |
| | Md | 9.00 | 9.50 | 11.00 | 10.50 |
| | Z | −1.75 | −1.30 | .40 | −.18 |
| | p | .04 | .10 | .34 | .43 |
| | r | .47 | .35 | .11 | .05 |
| Reading + Language + Attention N = 13 | M | 8.69 | 7.69 | 10.23 | 8.92 |
| | SD | 2.32 | 2.98 | 2.01 | 2.10 |
| | Md | 9.00 | 7.00 | 10.00 | 9.00 |
| | Z | −1.83 | −2.61 | 0.41 | −1.62 |
| | p | .04 | .004 | .34 | .05 |
| | r | .51 | .72 | .11 | .45 |

**Notes.**
Means (M), Standard Deviations (SD), Medians (Md), Wilcoxon Signed Rank Test Values (Z), P Values (p), and Effect Size Values (r) For Self-Concept Scores Produced By Our Sample Of Poor Readers (N = 77). r = 0.1, 0.3, and 0.5 Represent Small, Medium, And Large Effects Respectively.

(i.e., if they have poor spoken language), we used Spearman Rho Rank Correlation Coefficients to explore which types of reading ability (phonological recoding accuracy, visual word recognition accuracy, reading fluency, reading comprehension), spoken language ability (phonological processing, receptive vocabulary, expressive vocabulary), or attention (inattention, hyperactivity) might be reliably associated with these two types of self-concept. The coefficients are shown in Table 3 (Note that a similar table including correlation coefficients for home and social self-concept is included in Table A1 for interested readers). In line with *Cohen (1992)*, correlations of 0.1, 0.3, and 0.5 were considered small, moderate, and large, respectively. As outlined in the Introduction, we did not have clear predictions for the third step of the analysis, which was exploratory in nature. We therefore used a more conservative criterion for statistical significance the previous analyses, with $p < .05$. Statistically significant coefficients in Table 3 are marked in grey.

**Table 3** **Relationships between academic and general self-concept and reading, language, and attention.** Spearman Rho Rank Correlation Coefficients between academic and general self-concept and (1) Reading ability (phonological recoding accuracy, visual word recognition accuracy, reading fluency, and reading comprehension separately), (2) Spoken language ability (phonological processing and spoken vocabulary knowledge), and (3) Attention (inattention and hyperactivity). Statistically significant coefficients are marked in grey (two-tailed; $p < .05$).

| | Academic self-concept | | General self-concept | |
|---|---|---|---|---|
| | $r_s$ | $p$ | $r_s$ | $p$ |
| Phonological recoding | .24 | .04 | .01 | .94 |
| Visual word recognition | .33 | .004 | .20 | .09 |
| Reading fluency | .42 | <.001 | .22 | .05 |
| Reading comprehension | .29 | .01 | .29 | .01 |
| Phonological processing | .13 | .25 | .16 | .18 |
| Receptive vocabulary | .28 | .01 | .40 | <.001 |
| Expressive vocabulary | .10 | .37 | .26 | .02 |
| Inattention | −.22 | .06 | −.16 | .17 |
| Hyperactivity | .11 | .33 | −.06 | .63 |

The coefficients in Table 3 suggest four interesting trends: (1) most statistically significant correlation coefficients were moderate in size, with only two moderate-to-large in size; (2) neither academic nor general self-concept were reliable correlated with phonological processing, inattention, or hyperactivity in poor readers; (3) academic self-concept was reliably associated with all reading measures and one spoken language measure (receptive vocabulary); and (4) general self-concept correlated with measures that taxed receptive or expressive vocabulary - including the reading test most strongly correlated with vocabulary (i.e., reading comprehension: $r_s = .42, p < .001$) and expressive vocabulary ($r_s = .33, p = .003$). This suggests that general self-concept is related to spoken vocabulary rather than reading per se. In line with this suggestion, there was an almost non-existent relationship between general self-concept and phonological recoding ($r_s = .01, p = .94$), which itself had very weak relationships with receptive vocabulary ($r_s = .17, p = .15$ and expressive vocabulary ($r_s = .05, p = .66$).

## DISCUSSION

The aim of this study was to better understand *which types of poor readers have low self-concept.* We addressed this aim in three analyses. Below, we outline the outcomes of each analysis, and whether the outcomes supported our predictions based on existing evidence (if any).

### Analysis 1: prevalence of low academic, general, home, and social self-concept in poor readers

We estimated the prevalence of low scores in our poor readers for four types of self-concept (academic, general, home, and social) to test the prediction that poor readers would have a disproportionately high number of low scores for their academic self-concept but not their general, home, or social self-concept. Compared to a typical population, our poor

readers had significantly higher rates of low scores for academic self-concept and general self-concept but not home self-concept or social self-concept. This supports our prediction that academic self-concept would be low in poor readers, but rebuts our prediction that general self-concept, along with home and social self-concept, would be typical in poor readers. These findings support previous studies that have reported that poor readers have low academic self-concept (e.g., *Snowling, Muter & Carroll, 2007*; *Terras, Thompson & Minnis, 2009*); low general self-concept (*Riddick, 2009*); typical home self-concept (e.g., *Alexander-Passe, 2006*; *Westervelt et al., 1998*); and typical social self-concept (e.g., *Snowling, Muter & Carroll, 2007*; *Terras, Thompson & Minnis, 2009*). In doing so, our findings simultaneously fail to support a few previous studies that have found poor readers have typical academic self-concept (e.g., *Alexander-Passe, 2006*); typical general self-concept (*Westervelt et al., 1998*); low home self-concept (*Thomson & Hartley, 1980*); and low social self concept (*Boetsch, Green & Pennington, 1996*). Considered together, the weight of current evidence suggests that poor readers are at higher risk for academic and general self-concept but not home or social self-concept. It is noteworthy that this mixed support for different types of low self-concept in poor readers is conducive with the idea that not all poor readers have low self-concept.

## Analysis 2: comparing self-concept in poor readers with and without poor language and/or attention

We divided our poor readers into four groups (Reading, Reading + Language, Reading + Attention, Reading + Language + Attention) to test the prediction that poor readers with concomitant problems with spoken language or attention would have more severe problems with self-concept than poor readers with poor spoken language or poor attention, who would have more problems with self-concept than poor readers without problems with spoken language or attention. Again, the results partially supported our prediction. Children with poor reading alone did not have reliably low scores in any domain of self-concept. As predicted, children with either poor spoken language or poor attention had greater problems with self-concept than poor readers without these problems, with atypically low scores for their age for academic self-concept (Reading + Language, Reading + Attention) and general self-concept (Reading + Language group). Contrary to prediction, poor readers with both poor spoken language and poor attention did not have poorer self-concept than those either a problem with spoken language or attention. Indeed, they looked similar to children in the Reading + Language double deficit group, with atypically low scores for both academic self-concept and general self-concept. Thus, it was the presence of a concomitant deficit in language or attention that seemed to determine if academic self-concept was impaired in poor readers, and it was the presence of a spoken language problem that seemed to determine if general self-concept was impaired in poor readers.

These findings suggest, for the first time, that whether or not a poor reader has poor self-concept depends on whether they have a comorbid problem with language or attention, and that the type of self-concept problem that they have will depend on the type of comorbid problem that they have. This suggestion offers a potential explanation for the mixed outcomes of previous studies of self-concept in poor readers. To wit, whether or

not a particular study finds evidence for low self-concept in poor readers will depend on an interaction between the type of poor reader recruited for the study (e.g., with comorbid difficulties with language, comorbid difficulties with attention, no comorbid difficulties) and the type of self-concept tested by the study (e.g., academic self-concept, general self-concept, home self-concept). Unfortunately, it is difficult to test the validity of this explanation at this point in time because very few studies have assessed and reported the spoken language and attention abilities of participants. The notable exception is *Snowling, Muter & Carroll (2007)* who reported that their poor readers had verbal IQ scores (a broad index of spoken language ability) and attention abilities below the average range. In line with the outcomes of the current study, these poor readers had poor academic self-concept.

### Analysis 3: the relationship between poor readers' self-concept, reading, spoken language, and attention

We used correlation coefficients to determine which types of reading problems, spoken language problems, and attention problems, were reliably associated academic and general self-concept. As noted in the Introduction, such relationships have never been examined before within a group of poor readers, and hence no a priori predictions could be made. One key trend in the outcomes was that academic self-concept was reliably associated with multiple measures—specifically, all the reading measures and one language measure (receptive vocabulary). In line with previous research, this suggests that problems with reading (e.g., *Alexander-Passe, 2006*; *Terras, Thompson & Minnis, 2009*) or spoken language (*Carroll & Iles, 2006*; *Carroll et al., 2005*; *Lindsay & Dockrell, 2000*) may put a person at higher risk for low academic self-concept. The outcomes of the current study further suggest that this risk may only reach significant levels in poor readers who have a comorbid deficit in another cognitive domain such as language.

Another interesting trend in the coefficients was that general self-concept only correlated with measures that taxed receptive or expressive vocabulary to some degree, including reading comprehension, reading fluency, and visual word recognition. Thus, poorer receptive or expressive vocabulary skills may put a child at higher risk for low general self-concept, but poor reading ability and poor attention do not.

### Theoretical implications

Understanding which types of poor readers have low self-concept is useful for developing a more complete theory about why there might be association between poor reading and poor self-concept. At this point in time, theoretical explanations about the mechanisms linking poor reading and low self-concept are underspecified. For example, it has been proposed that (1) poor reading leads to low self-concept, (2) low self-concept leads to poor reading, (3) a third factor causes both poor reading and low self-concept, or (4) there is a two-way interaction in which poor reading causes low self-concept and then low self-concept in turn causes poor reading (e.g., *Battle, 2002*; *Riddick, 2009*). These theoretical accounts do not identify the mechanisms that might form a causal chain linking poor reading to low self-concept (or vice versa).

The outcomes of this study make four tentative suggestions about the nature of these mechanisms. First, finding only modest correlation coefficients between reading and certain domains of self-concept (academic and general) could be taken to suggest that the link between poor reading and poor self-concept is not a close nor direct one. Instead the causal chain linking reading to self-concept, if one exists, may comprise numerous links, each of which may moderate (i.e., reduce) the overall strength of the relationship between the end points of the chain (i.e., reading and self-concept). Second, finding that children with poor reading alone had no problems with self-concept might indicate that poor reading per se may not be enough to trigger a chain of causal events leading to poor academic self-esteem. Third, finding that children with poor reading plus poor language or poor attention have poor academic self-concept raises the possibility that academic self-concept is only at risk when poor reading is paired with another cognitive deficit. Fourth, finding that poor readers with poor spoken language have poor general self-concept suggests that spoken language impairment may have a relationship with general self-concept that is independent of reading impairment.

Considering these four possibilities together, we cautiously hypothesise that if a child "only" has a reading impairment, they may not be at risk for low self-concept because their parents and friends view their specific reading impairment as a unique exception to their child's overall academic, general, home, or social abilities. This may minimize negative feedback that a child receives about her- or himself, and hence preserve their self-concept in all domains. However, if a child has poor reading in conjunction with another deficit— such as a problem with spoken language or attention, or perhaps mathematics, writing, or motor co-ordination—their parents and friends may form a negative view about that child's academic or general ability, and hence the child may receive negative feedback from significant others about their ability to succeed at school or in life in general. This may negatively affect their own perception of their academic or general abilities, and hence result in lower academic or general self-concept. The idea that children with multiple impairments are at higher risk for low self-concept is consistent with cumulative risk models of developmental disorders which suggest that while a single risk factor (e.g., poor reading) may increase a child's propensity for a cognitive, environmental, socio-emotional or physical health problem, this impairment may only reach a significant or clinical level when that single risk factor is paired with at least one other risk factor (*Aro et al., 2009*; *Dilnot et al., 2016*).

## Clinical implications

The outcomes of the current study hint at how poor readers' self-concept might be assessed and treated in clinical practice. First and foremost, the results suggest that poor readers be tested for their self-concept in the academic and general domains. Given the individual differences in self-concept discovered in this study, one cannot predict if a poor reader is at risk for low self-concept without testing them explicitly.

If, for some reason, it is not possible to get a reliable index of a poor readers' self-concept—because they are too young, or time and money is limited—a clinician might consider using a child's developmental history to predict if they might be risk of low

self-concept. If a child has a history of delayed language development or poor attention, the extra time and expense involved in testing academic and general self-concept may be justifiable. However, if a child with poor reading has no history of delayed language or inattention, they may be considered at low risk for any type of low self-concept, and hence no further testing may be required.

This latter suggestion highlights a positive finding of this study: Children who have poor reading but no concomitant problems in spoken language or attention do not appear to be at risk for low self-concept. This outcome was not predicted by existing evidence, probably because few previous studies have tested poor readers for their spoken language or attention. Yet again, the exception is *Snowling, Muter & Carroll (2007)*, who reported that their sample of children with poor reading, whose spoken language scores were below the average range as a group, had low academic self-concept but not low social or athletic self-concept.

## Limitations

The outcomes of this study, along with its potential theoretical and clinical implications, must be considered within the context of its methodological strengths and weaknesses. The vast majority of previous studies of self-concept in poor readers tested fewer than 50 poor readers. Thus, the current study joins a relatively small group of studies with a relatively large sample (*Polychroni, Koukoura & Anagnostou, 2006*; *Lau & Chan, 2003*; *Boetsch, Green & Pennington, 1996*; *Maughan & Hagell, 1996*; *Murray, 1978*). However, the division of our 77 poor readers into smaller subgroups in the second analysis would have reduced the power of this particular analysis. Effect-size estimates in Table 2 suggest that the lower power of this analysis had minimal impact on the main findings, since all large effects ($N = 6$) were statistically significant and all small effects ($N = 7$) were non-significant. It was only a few moderate effects ($N = 3$) that fluctuated in significance, one being significant in a larger group ($N = 25$), and two being non-significant in the smaller groups ($N = 13$ and 14). It would be helpful if a future study could clarify the reliability of these few moderate findings using larger groups of poor readers with and without spoken vocabulary and inattention for their different types of low self-concept.

Another potential limitation of this study was the use of a single psychometric measure to assess different types of self-concept. At this early stage of research, we felt that this was important to ensure that the different scales of self-concept were comparable in terms of norms and reliability and validity. Now we have found that self-concept differs between different types of poor readers, it would be helpful if new, well-powered, studies could use alternative tests of academic self-concept and general self-concept to determine if our findings are replicable with other measures of self-concept.

A third limitation of this study is that it was a correlational study rather than a causal study. A correlational approach is useful for starting to develop a theory about which mechanisms might form a causal chain linking poor reading to low self-concept, and for starting to make predictions in clinical practice about whether a poor reader is at risk for low self-concept, and what type of low self-concept they might have. However, a correlational study cannot tell us the direction of causal effects in a theoretical chain of cognitive factors

linking reading and self-concept, and it cannot tell us if clinical practice should treat poor reading to improve self-concept, to treat self-concept to improve poor reading, or to treat both simultaneously. It would therefore be extremely useful if a randomised control trial compared three types of treatment (i.e., reading, self-concept, reading and self-concept) in poor readers with poor spoken vocabulary and poor readers with inattention.

## SUMMARY

The aim of this study was to better understand which types of poor readers have low self-concept. We tested 77 children with poor reading for different types of reading, spoken language, and attention. We discovered that poor readers had disproportionately high number of low scores for their academic and general self-concept but not their home or social self-concept; that poor readers with poor attention had low academic self-concept; that poor readers with poor spoken language had low general self-concept as well as low academic self-concept. These findings have both theoretical and clinical implications, and encourage further investigations into the heterogeneous nature of self-concept in poor readers.

## ACKNOWLEDGEMENTS

We would like to thank all the children and parents who donated their time and effort to this research; the team of research assistants who helped collecte the data for this project (Kristy Jones, Linda Larsen, Thushara Anandakumar, Huachen Wang, Pip Eve, Kate Glenn); and the editor and reviewers for their valuable contributions to the development of this manuscript.

## APPENDIX

**Table A1** **Relationships between home and social self-concept and reading, language, and attention.** Spearman Rho Rank Correlation Coefficients between home and social self-concept and reading ability (phonological recoding accuracy, visual word recognition accuracy, reading fluency, and reading comprehension separately), Spoken language ability (phonological processing and spoken vocabulary knowledge), and Attention (inattention and hyperactivity). Statistically significant coefficients are marked in grey (two-tailed; $p < .05$).

| | Home self-concept | | Social self-concept | |
|---|---|---|---|---|
| | $r_s$ | $p$ | $r_s$ | $p$ |
| Phonological recoding | .06 | .63 | .16 | .17 |
| Visual word recognition | .08 | .47 | .05 | .67 |
| Reading fluency | .25 | .03 | .19 | .10 |
| Reading comprehension | .19 | .09 | .24 | .03 |
| Phonological processing | .10 | .39 | .09 | .44 |
| Receptive vocabulary | .30 | .008 | .30 | .009 |
| Expressive vocabulary | .12 | .29 | .18 | .11 |
| Inattention | −.19 | .11 | −.17 | .15 |
| Hyperactivity | −.02 | .79 | −.17 | .14 |

### Funding

This research was funded by NHMRC Project 488518 and ARC DP0879556. The funders had no role in study design, data collection and analysis, decision to publish, or preparation of the manuscript.

### Grant Disclosures

The following grant information was disclosed by the authors:
NHMRC Project: 488518.
ARC: DP0879556.

### Competing Interests

Professor McArthur is an Academic Editor of PeerJ.

### Author Contributions

- Genevieve McArthur conceived and designed the experiments, performed the experiments, analyzed the data, contributed reagents/materials/analysis tools, wrote the paper, prepared figures and/or tables, reviewed drafts of the paper, developed cognitive tests and obtained funding.
- Anne Castles conceived and designed the experiments, performed the experiments, contributed reagents/materials/analysis tools, wrote the paper, reviewed drafts of the paper, developed cognitive tests and obtained funding.
- Saskia Kohnen conceived and designed the experiments, performed the experiments, contributed reagents/materials/analysis tools, reviewed drafts of the paper, developed cognitive tests.
- Erin Banales conceived and designed the experiments, performed the experiments, contributed reagents/materials/analysis tools, reviewed drafts of the paper, lead research assistant team.

### Human Ethics

The following information was supplied relating to ethical approvals (i.e., approving body and any reference numbers):

Macquarie University Human Research Ethics Committee, approval number: 5201200852.

### Data Availability

The data for this study was collected before open source data repositories became common, thus we did not ask (and hence do not have consent) from each participant for their data to be included in an open-source repository. Our ethics committee confirmed we were able to provide source data to reviewers, but interested readers would need to be added to our ethics approval in order to access the data for further analysis. Please submit your requests to the corresponding author: Genevieve McArthur, genevieve.mcarthur@mq.edu.au.

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
