# Peer review of "Low self-concept in poor readers: prevalence, heterogeneity, and risk"

_PeerJ, doi:10.7717/peerj.2669_

## Round 0.1 · original submission · Minor Revisions

The Reviewers have raised a number of questions, and made suggestions and useful comments that would make this a stronger submission. Please read their comments, and make appropriate changes to your paper. In particular, there is a need to clarify your approach to data analysis, provide more information on how assessments were made, and explain why a liberal p-value is used with the one-tailed tests. I agree that this is somewhat questionable, particularly as there appears to be no need. Please respond to the questions raised by the Reviewers, and correct the inappropriate use of causal language, as well as the small errors, and numerical formatting. Avoid over-stating the results.

·

Basic reporting

This article begins with a review of the literature on poor readers and self-concept generally. This goes on to explore the different types of self-concept that poor readers may have increased risk within. It may be more useful to explore the different types of reading difficulties that poor readers have, prior to the discussion of self-concept, as it is not clear what is meant by being a poor reader until the end of the study. The first paragraph does briefly explore poor readers but it is not explored again until the end of the section.
The article is well structured and appropriately referenced. The overall research stands alone as a single piece of research and all elements come together as a coherent whole.

There are one or two small errors:
• The first word on line 175 should be complete, not completed
• Line 296-298 says: Outcomes of the first analysis suggested that distributions of scores for at least two of the self-concept measures (academic and general) might not differ significantly from a normal distribution. The addition of ‘not’ here must be wrong as these two measures didn’t match the expected normal distribution, hence the use of the non-parametric tests.
• Line 403 missing the word home when referring to home self-concept
• A word missing on line 429

Experimental design

This article describes original primary research, namely the exploration of the distribution of self-concept issues across poor readers with and without comorbid problems that may be related to the act of learning to read. It is a limitation of the study that it is correlational rather than causal, and this is noted by the authors themselves. However, overall the design itself is appropriate for the exploratory nature of the research aim and specific predictions. The three research predictions, as they and their relationship to the analyses were clear and coherent.

Two queries in the measures section:
• Both the phonological processing and receptive vocabulary paragraphs start with the statement, ‘in line with … Bishop and Snowling’… but the previous reference to Bishop and Snowling was 6 pages earlier. It might need a statement to restate the purpose, although the distinction between the two aspects of language doesn’t seem controversial enough to argue for it when describing the measures.
• The attention measures mention clinical significance, with the term in quotation marks for inattention and not for hyperactivity. Was there a reason for this?

Validity of the findings

The conclusions are clearly linked to the research predictions and the analyses carried out. The overall finding was that poor reading comorbid with language contributes to a greater risk for low self-concept generally as well as academically. Poor reading comorbid with poor attention contributes to risk for low academic self-concept. The theoretical and clinical implications are both appropriate to the findings.
The discussion regarding negative feedback was more subjective, but not inappropriate. It may of interest in the future to examine difficulties across domain, for example in mathematics as well as reading as children recognising they have difficulties in more than one domain, as compared to their peers, would negatively affect their own perception more than negative feedback from others.

Reviewer 2 ·

Basic reporting

There are a few instances where the authors use numerals in a way that is confusing. For example, p.2 line 66 states that 26 8 to 12 year olds… formatting suggestion would be twenty-six 8 to 12 year olds…

An additional paragraph in the introduction describing problems and/or issues associated with low self-concept might provide a stronger rationale regarding the importance of this study.

This is also a fair amount of relevant research on the domain specificity of self-concept that was not covered in the current introduction that seems particularly relevant to the aims of the study.

Overall, the introduction provides a sound rationale for the current study.

Experimental design

While the list of measures was comprehensive and appeared appropriate for the goals of the study, very little information was provided on how these assessments were administered. There were quite a few and, given that these were done with relatively young children, it would be important to understand how assessment sessions were structured/scaffold to support the collection of high quality data.

Given the large number of variables, it would also be helpful to have a brief section describing the approach to data analysis and rationale for why these approaches were selected prior to reporting Results.

Validity of the findings

The authors used a very liberal criterion for statistical significance by employing one-tailed tests (which was supported by a sound rationale) and applying a critical p-value of .10, which was more questionable. Given the results were sig at the .01 level, it is unclear why such a liberal criteria was needed.

The use of non-parametric tests was appropriate.

Comparisons of the 4 groups on reading measures might have been non-significant due to low power, given that groups sizes were in the range of 13-14 children and no larger than 25. Reporting of effect sizes might be useful here to provide additional evidence for consideration on these comparisons.

Given that the one-tailed tests were selected, the brief discussion of the higher scores for the reading group should be removed.

The findings from the current study are not fully explained/interpreted in relation to findings from previous (related) studies. For example, very little is offered in terms of why there appear to be inconsistent results across studies examining the association between reading skill and various types of self-concept.

The authors should be careful using language regarding causal links and moderation, such as was used on line 470 of the discussion. These are all testable/empirical questions, some of which could be examined using the data that were collected. If the authors believe some sort of moderation or mediation is present, they should test these hypotheses explicitly.

There is a mention of “solid power”, yet no power analysis was ever reported. A sample of 77 seems generally adequate for the analyses used (t-tests and correlations); however, limitations to power with a sample of this size might have impacted the results when the data were split into four groups. Thus, the statements regarding power should be supported and might be overstated.

In general, the implications seemed to outreach the data/Results and could be tempered.

Additional comments

Overall the study was well written and interesting to read.

---

## Round 0.2 · Minor Revisions

The Revision has addressed all of the comments and suggestions raised by the Reviewers and is much improved. I think it is an important addition to our knowledge about low self-concept and poor reading.

There is one point that I want to raise in your revision. With regard to the description of your sample on pg 3, what exactly is inappropriate reading instruction (instructional casualties)? Do you know what type of reading instruction the children were given? This is not mentioned in the Participants section. Although a minor point it is an important one having both theoretical and clinical implications.

---

## Round 0.3 · accepted · Accept

I agree a study on self-concept in poor readers to see if it differed between groups of children who received different types of instruction would be most interesting.